# ▽ SpiritSight Agent:
# Advanced GUI Agent with One Look

## Abstract

Graphical User Interface (GUI) Agents show amazing abilities in assisting human-computer interaction, automating human user's navigation on digital devices. An ideal GUI Agent is expected to achieve high accuracy, low latency, and generality across various GUI platforms. Recent visual-based approaches show promises, taking the advantages of advanced Vision Language Models (VLMs). Although they generally meet the requirements of generality and low latency, these visual-based GUI Agents often fall short in terms of localization accuracy. To address this issue, we propose **SpiritSight**, a visual-based generalist end-to-end GUI agent with outstanding grounding abilities. First, we create a multi-level, large-scale, high-quality GUI training dataset with scalable methods and train SpiritSight using curriculum learning, empowering it with robust GUI understanding and localization capabilities. Second, we introduce the **Universal Block Parsing (UBP)** method, which frames the localization task as a multi-image QA problem, further enhancing SpiritSight's ability to ground GUI objects. With the above-mentioned efforts, SpiritSight constantly outperforms previous SOTA methods across numerous major automatic GUI navigation benchmarks. Notably, SpiritSight-8B achieves a 46.1% Step Success Rate(Step SR) on the Mind2Web benchmark without any candidates element input, **more than doubling** the performance of SeeClick (20.9%) with a comparable model scale. SpiritSight also outperforms other visual-language-based methods in various GUI platforms, demonstrating its superior capability and compatibility in GUI Agent tasks. The models and the code will be made available upon publications.

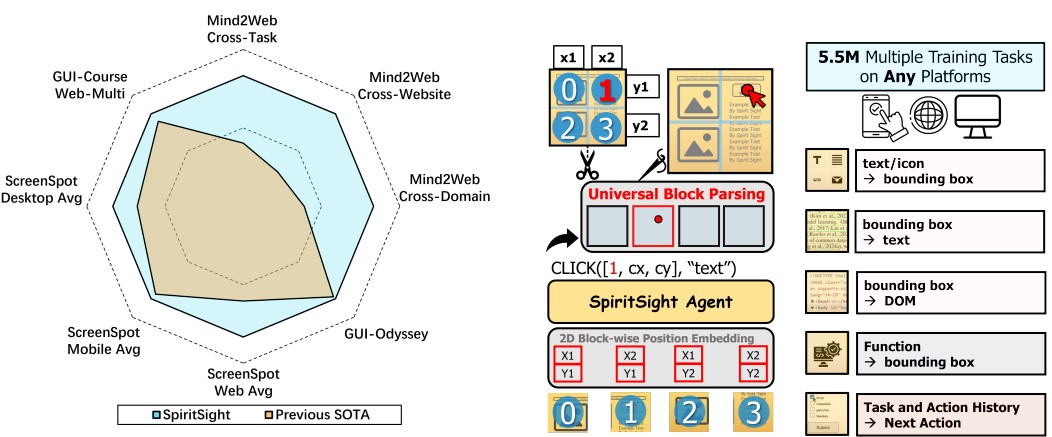

(a) The performance of SpiritSight Agent in comparison with previous SOTA approaches.

(b) An overview of SpiritSight Agent's solution.

Figure 1: (a) Our model achieves new state-of-the-art (SOTA) performance across benchmarks in web, mobile, and desktop scenarios. (b) We introduced the Universal Block Parsing (UBP) method, which replaces the global coordinate representation with a relative coordinate for each block sub-image, significantly enhancing the model's grounding capabilities. We also developed a large-scale curriculum learning dataset that equips models with three levels of comprehensive GUI knowledge.

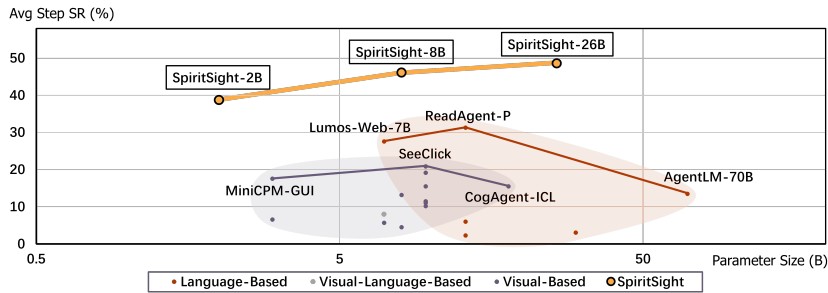

Figure 2: Comparison of the Average Step SR on Mind2Web benchmark of our SpiritSight Agent of three sizes(2B, 8B, 26B) with various previous approaches. We constantly surpass all of them.

# 1 INTRODUCTION

Graphical User Interface (GUI) automation has long been pursued by people along with the development of the modern digital devices. Thanks to recent advances of Large Langue Models (LLMs), GUI Agents are constructed to assist users in interacting with graphical interfaces, automatically making action decisions based on observations of environmental elements and user's objective.

Current approaches can be divided into three categories based on their inputs. Language-based and visual-based approaches make use of Hyper Text Markup Language (HTML) / Extensible Markup Language (XML) and screenshots as input (Zheng et al., 2023; Huq et al., 2023; Deng et al., 2024; Wan et al., 2024; Lai et al., 2024; Lee et al., 2024; Yin et al., 2024; Hong et al., 2024; Cheng et al., 2024; Chen et al., 2024b), respectively. Visual-language-based methods integrate multi-modal information by enhancing HTML with screenshots (Furuta et al., 2023; Thil et al., 2024; Kil et al., 2024; Zheng et al., 2024).

The language-based and visual-language-based methods typically applied only in the web domain, and often limited by the excessive length of HTML or security concerns regarding it. The visual-based approaches demonstrates enhanced compatibility across various GUI platforms, as acquiring screenshots is generally easier than obtaining hierarchical data from platforms except for the web. However, visual-based approaches struggle to localize the elements objects (*i.e.* buttons, text boxes) from the input visual context. Some works solve this problem by adopting Dynamic High-Resolution (Kim et al., 2022; Chen et al., 2024c) approach, which may bring ambiguity to the process of model learning. Others attempt to collect large scale training data through manual synthesis (Shi et al., 2017; Liu et al., 2018; Lee et al., 2023), human annotation (Yao et al., 2022; Deng et al., 2024; Rawles et al., 2024; Chen et al., 2024a; Chai et al., 2024; Lu et al., 2024; Lù et al., 2024) and the use of common datasets (Deka et al., 2017; Li et al., 2020; Wang et al., 2021; Cheng et al., 2024; Zhang et al., 2024a), while these data are respectively unrealistic, expensive and of low quality.

To address the aforementioned challenges, in this paper, we proposed a single-stage, visual-based GUI Agent——SpiritSight, which has strong ability in GUI navigation task. Our contributions are summarized as follows.

**Firstly, we propose a cost-effective GUI dataset of 5.46 million samples to enhance our model's GUI understanding and localization capabilities.** The datasets is collected from real-world and filtered through carefully designed rules to ensure data quality. They are also constructed with a clear hierarchy and consist of 3 different level of components: text/icon recognition and grounding tasks, functional grounding task, and GUI navigation task. The first two parts of datasets, which constitute 90% of the total and have been collected for free, are primarily used to equip our model with robust elements grounding capabilities, thereby improve its GUI navigation ability.

**Secondly, We introduce a Universal Block Parsing (UBP) method to resolve the ambiguity in Dynamic High-Resolution input.** This method treats the localization task as a multi-image QA problem (Raj et al., 2021), where each element object are grounded within the corresponding sub-image. It also introduce a 2-dimensional block-wise position embedding method (Kim et al.,

2022) to help the model learn the spacial information of cropped input image, thereby enhances the grounding capabilities of SpiritSight.

**Thirdly, we evaluate our SpiritSight model family in various GUI benchmark and it exhibits impressive performance among them.** We release two versions of GUI Agent with different model size: the large-scale SpiritSight-26B, standard SpiritSight-8B and the lightweight SpiritSight-2B. SpiritSight-2B achieve a 96% hit rate on text/icon grounding task, demonstrating near-perfect performance in pure grounding tasks. On the ScreenSpot (Cheng et al., 2024) benchmark, SpiritSight-8B achieve a 66.5% accuracy and surpasses SeeClick (8B) (Cheng et al., 2024) by 13.1%, and by 19.1% over CogAgent (18B) (Hong et al., 2024). Under the non-candidate input setting, SpiritSight-8B and SpiritSight-26B attains an average Step Success Rate of 46.1% and 50.2% on the Mind2Web (Deng et al., 2024) benchmark, outperform all works including language-based, visual-based and even visual-language-based methods.

## 2 RELATED WORK

### 2.1 LANGUAGE-BASED AND VISUAL-LANGUAGE-BASED GUI AGENT

Several works leverage the capabilities of Large-scale Language models (LLMs) to construct GUI agents. It is noticed that they are mostly multi-stage architectures. Mind2Web (Deng et al., 2024) employs a lightweight language model to extract candidate elements from HTML, followed by a ranking model that sorts the elements based on task descriptions and historical actions. Finally, a large language model predicts actions and the elements on which they are applied. WebAgent (Gur et al., 2023) first uses an encoder-decoder model to generate low-level instructions and relevant HTML code snippets, then uses another decoder to produce executable Python code. AutoWebGLM (Lai et al., 2024) simplifies HTML code through manually designed rules before predicting the action codes.

Other visual-language-based works leverage both GUI screenshots and hierarchical HTML/XML to enhance the robustness of GUI agents. WebGUM (Furuta et al., 2023), CC-Net (Thil et al., 2024) use ResNet and ViT to extract features from screenshots respectively. The image embedding are then combined with text embedding and fed into a multi-modal transformer. SeeAct (Zheng et al., 2024), AppAgent (Yang et al., 2023) identify all interactive elements using HTML files or XML files. It then assign each interactive element a unique identifier in the screenshot and then feed the screenshot into the model.

These language-based methods or visual-language-based methods that rely on the hierarchical information exhibit several limitations: (1) Acquiring hierarchical representations like HTML/XML are not equally available on different platforms. And even this information is available, their internal rule difference makes language-based GUI Agents less compatible; (2) HTML often contains redundant and customized information, requiring additional models or extensive manually crafted rules for effective filtering.(3) Text-based GUI Agents are vulnerable to injection attacks (Zhan et al., 2024; Wu et al., 2024; Liao et al., 2024), where malicious instructions hidden in HTML can easily lead to erroneous or unsafe actions.

### 2.2 VISUAL-BASED GUI AGENT

Recently, some visual-based approaches have been proposed to overcome the drawbacks of language-based methods. Some of them (Shaw et al., 2023; Hong et al., 2024; Cheng et al., 2024; Baechler et al., 2024) are single-stage methods that only use GUI screenshots as input for MLLMs and output the next action in an end-to-end manner. However, these agents perform worse on relevant GUI benchmarks compared to other approaches. MobileAgent and MobileAgent-v2 (Wang et al., 2024b;a) are two-stage methods, using the GPT-4V API instead of publicly available MLLMs. They find that top models like GPT-4V are not adept at element grounding tasks, thus introduce additional tools such as OCR and icon recognition models to assist with localization. However, this may increase the complexity and inference latency of the agent system. Overall, the current MLLMs demonstrate poor localization capabilities for GUI grounding task, limiting the navigation capabilities of single-stage visual-based GUI agents. In Appendix A, We discuss additional related works on large-scale language Models (LLMs) and Multi-modal Large Language Models (MLLMs).

Table 1: Results of SpiritSight on Mind2Web Benchmark. [*] indicates that this model select from the top-50 candidate elements (Lai et al., 2024; Hong et al., 2024; Zheng et al., 2024). [††] indicates visual-language-based methods (Zheng et al., 2024), while [†] indicates language-based methods (Lee et al., 2024). Others are all visual-based methods. (Chen et al., 2024b; Bavishi et al., 2023; Cheng et al., 2024)

| | Model | Cross-Task | | | Cross-Website | | | Cross-Domain | | |
|---|---|---|---|---|---|---|---|---|---|---|
| | Size | Ele.Acc | Op.F1 | Step SR | Ele.Acc | Op.F1 | Step SR | Ele.Acc | Op.F1 | Step SR |
| HTML-T5-XL[*] | 3B | - | - | 71.5% | - | - | 62.2% | - | - | 67.1% |
| AutoWebGLM[*] | 6B | - | - | 66.4% | - | - | 56.4% | - | - | 55.8% |
| LLaMA2-7B[*] | 7B | - | - | 52.7% | - | - | 47.1% | - | - | 50.3% |
| CogAgent[*] | 18B | - | - | 62.3% | - | - | 54.0% | - | - | 59.4% |
| SeeAct[††] | - | 46.4% | 73.4% | 40.2% | 38.0% | 67.8% | 32.4% | 42.4% | 69.3% | 36.8% |
| ReadAgent-P[†] | 340B | 33.7% | 72.5% | 29.2% | 37.4% | 75.1% | 31.1% | 37.2% | 76.3% | 33.4% |
| MiniCPM-GUI | 3B | 23.8% | 86.8% | 20.8% | 20.3% | 81.7% | 17.3% | 17.9% | 74.5% | 14.6% |
| Fuyu-GUI | 8B | 19.1% | 86.1% | 15.6% | 13.9% | 80.7% | 12.2% | 14.2% | 83.1% | 11.7% |
| SeeClick | 9.6B | 28.3% | 87.0% | 25.5% | 21.4% | 80.6% | 16.4% | 23.2% | 84.8% | 20.8% |
| SpiritSight-2B | 2B | 51.7% | 87.2% | 44.9% | 44.0% | 83.6% | 37.8% | 42.4% | 83.5% | 36.9% |
| SpiritSight-8B | 8B | 59.2% | 88.9% | 52.7% | 52.2% | 84.7% | 44.0% | 50.1% | 86.0% | 44.4% |
| SpiritSight-26B | 26B | **60.5%** | **89.7%** | **54.7%** | **57.0%** | **85.7%** | **48.1%** | **54.1%** | **87.2%** | **49.2%** |

## 3 DATA COLLECTION

In this chapter, We introduce a data collection strategy specifically designed to address the deficiency in a visual-base GUI agent. We highlight the deficiency by modeling the GUI navigation task using a sequential decision-making process and further breaking it down through hierarchical policy decomposition. See Appendix B for details.

### 3.1 LEVEL ONE: VISUAL-TEXT ALIGNMENT

Visual text alignment refers to the model's ability to recognize or locate the text content of a text element or the icon caption of a icon element, which requires the source data of the GUI platform. On the web scenario, We collected website URLs from two sources: the common crawl (Group, 2024) datasets and URLs from website ranking. We then developed a data collection tool using playwright (Microsoft, 2024) library to get real-world web data from the collected URLs. With this tool, we collected 740k web-page screenshots along with their DOM annotation, with the diversity both in resolution and languages. We also noticed that the icons on the web-pages often lack captions. The existing icon detection (Bai et al., 2021; He et al., 2021) tools are not fully adaptable to the web scenario, due to their extensive use of custom-designed icons. So, we developed a InternVL-Icon as the icon annotation tool by collecting a dataset of 30K icon-caption pairs from Alibaba (2024) and fine-tuning InternVL1.5-26B using this dataset. After that, we annotate all the icons on web-pages by the captions generated from InternVL-Icon. As for the mobile scenario, we collect data from AitW (Rawles et al., 2024), which contains a large-scale GUI data in mobile devices.

Based on our collected source data, we construct three tasks: **text2bbox**, **bbox2text**, and **bbox2dom**. **Text2bbox** task prompts the model to ground the element based on the given text or icon caption. We additionally include context information for the elements that appears multiple times in a screenshot to avoid ambiguity. The text2bbox data is the most abundant among the three tasks, in order to help the model learn grounding capabilities. **Bbox2text** task is the inverse version of the text2bbox task, teaching the model about Optical Character Recognition (OCR) and icon captioning. **Bbox2dom** task requires the model to generate the DOM-tree based on the given bounding box area, as show in Figure 7. The bbox2dom is constructed to help model learns about the GUI layout knowledge besides the basic OCR and icon recognition. To make sufficient use of the context length of the model, We pack dozens of data pairs in one training sample for text2bbox and bbox2text task, and select the box that include as many elements as possible for bbox2dom task. Overall, we totally construct **1.9M** and **1.1M** training samples on web and mobile platforms. See Appendix F for more

details. These training data together largely enhance the GUI foundational abilities, especially the GUI grounding ability, of our SpiritSight model.

## 3.2 LEVEL TWO: VISUAL-FUNCTION ALIGNMENT

Visual-Function Alignment refers to the model's ability to recognize or locate the function of a element, where the element function data cannot be directly obtained from the real-world environment like in the first level. Inspired by the back-translation (Sennrich, 2015) method for data construction, who collect the dataset for the forward translation task using back-translation, we leverage InternVL's capabilities in image understanding to collect element function data.

We conducted custom tests on InternVL2-26B to evaluate its ability to recognize element functions before collecting data. We divided the screenshot into a 3x3 grid to represent the approximate location of the element (*i.e.* in the top-left corner of the image) and placed a bounding box around the target element in the screenshot to assist with specifying the element. By providing the model with the screenshot, the element's text content or icon caption, the region where the element is located, we prompted the model to generate the corresponding function of the element. Additionally, We utilize InternVL2.5-20B to enhance the quality and diversity of the generated function descriptions. InternVL2-26B achieving an 80% accept rate with human judgement, which we consider acceptable for constructing the functional grounding data.

Based on the methodology above, we collect element-to-function pairs for all the operable elements collected in level 3.1 and then reverse it to function-to-element pairs. Combined with the position annotations of the elements, we ultimately obtain **function2bbox** pairs. We use the same packing method as in the text2bbox and bbox2text data for efficient model training and ultimately obtained **0.9M** training samples. Besides, we also collect the functional grounding data for the mobile scenario, which is derived from the construction of the GUI navigation data, as described in the level 3 section.

## 3.3 LEVEL THREE: VISUAL GUI NAVIGATION

We utilize the public available AitW (Rawles et al., 2024) dataset to construct our GUI navigation training data. As mentioned in Zhang et al. (2024a); Chai et al. (2024), AitW data involves a certain amount of incorrectly labeled samples, so we decide to clean it with GPT-4o. We adopt the Chain-of-Thought (CoT) (Wei et al., 2022b) to make the judgment more accurate. Specifically, We prompt GPT-4o with the task objective, the screenshot at the current step and the next step, the previous actions, and the labeled current action. GPT-4o is required to first summarize the two screenshots and tell the difference between them, then describe the current actual step description according to the difference, and lastly assess the reasonability of current action. We filter out data samples deemed unreasonable and ultimately got **0.63M** GUI navigation samples.

With the collected CoT-style data, we are able to collect functional grounding data for mobile scenario that is mentioned in section 3.2, as each step includes a description. The collected data also allow us to train the model in a CoT manner to make it stable for model to learn and easy to converge.

## 3.4 OTHER TRAINING DATA

To enhance the model's understanding of GUI content, we further collected some public datasets as a supplement, including doc/web/mobile VQA datasets (Mathew et al., 2021; Chen et al., 2024b; 2021; Hsiao et al., 2022), image captioning datasets (Deka et al., 2017; Wang et al., 2021), and mobile grounding datasets (Li et al., 2020; Deka et al., 2017). Finally, we construct **0.49M** QA pairs from the datasets above.

# 4 UNIVERSAL BLOCK PARSING

## 4.1 PROBLEM STATEMENT

We build our model based on the pre-trained InternVL2.0 (Kim et al., 2022; Chen et al., 2024c), known as InternVL for short, a family of advanced and open-sourced VLMs. Its dynamic resolution

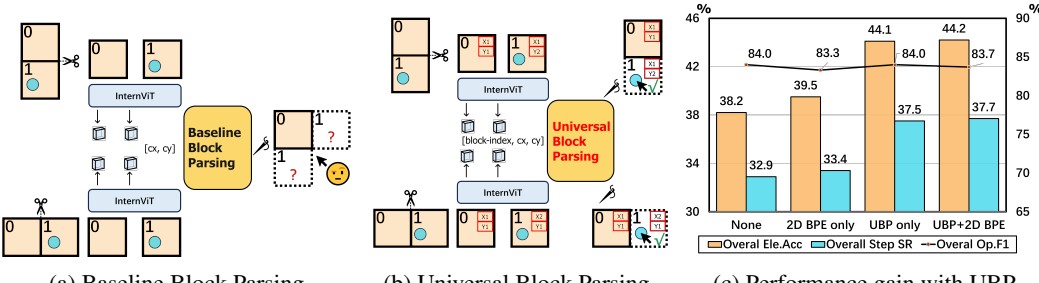

(a) Baseline Block Parsing.     (b) Universal Block Parsing.     (c) Performance gain with UBP.

Figure 3: (a) The Baseline Block Parsing method is used by previous works that uses a global coordinate system for the whole input image. **VS** (b) Our proposed Universal Block Parsing (UBP) that replace the global coordinates with relative ones that are specific to the block. (c) Comparison on Mind2Web benchmark for 2D Block-wise Position Embedding(2D-BPE), Universal Block Parsing (UBP), and the combination of 2D-BPE and UBP.

strategy largely preserves the details of the input screenshots by divided them into an optimal number of blocks. However, the dynamic resolution strategy may introduce problem in grounding GUI element.

As represented in Figure 3a and Figure 3b, To highlight the issue, we assume two input screenshots with aspect ratios of 1:2 and 2:1, respectively. In each screenshot, there is a target element, both of which are located in the same position within block-1 after the image cropping process. This leads to the model being expected to predict different locations during training for two samples in the same position, which we refer to as ambiguity.

## 4.2 METHOD

One solution is to input an additional thumbnail, but this may lead to extra computational and memory overhead. We propose to solve this positional ambiguity with two steps. Firstly, we introduce 2D Block-wise Position Embedding(2D-BPE)(Kim et al., 2022) by adding two position embedding to the sub-image feature. Secondly, we introduce a Universal Block Parsing (UBP) method, where we replace the global coordinates with relative ones that are specific to the block. Specifically, a point is expressed in global coordinate as

$$loc = [cx, cy] \tag{1}$$

Where $cx$ and $cy$ represent the horizontal and vertical coordinates values of the point in the original image, respectively. In the UBP method, we expresses the same point as the following derivation.

$$\begin{cases} w_{block} = \lceil \frac{w_{img}}{n_w} \rceil \\ h_{block} = \lceil \frac{h_{img}}{n_h} \rceil \end{cases} \quad \begin{cases} b_x = \lfloor \frac{cx}{w_{block}} \rfloor \\ b_y = \lfloor \frac{cy}{h_{block}} \rfloor \end{cases} \quad \begin{cases} b_i = b_y \cdot w_{block} + b_x \\ cx' = cx \mod block_w \\ cy' = cy \mod block_h \end{cases}$$

$$loc = [b_i, cx', cy'] \tag{2}$$

Where $w_{img}$ and $h_{img}$ represent the width and height of the original image, $n_w$ and $n_h$ represent the number of blocks in the columns and rows, respectively, and $b_i$ represent the block index. During the model inference, the global coordinate of this point can be parsed inversely by

$$\begin{cases} cx = cx' + (b_i \mod n_w) \cdot w_{block} \\ cy = cy' + \lfloor \frac{b_i}{n_w} \rfloor \cdot h_{block} \end{cases}$$

We assume that most GUI elements are small enough to be fully contained within a single block, rather than being split across multiple blocks. As a result, for most element objects, the single-image grounding task becomes a multi-image grounding task. For elements that are split between blocks, we assign their block index based on the location of the element's center, as described in Equation 2. This special case further improves the model's ability to understand spatial relationships between blocks, as it trains the model to restore the occluded parts. Overall, our UBP method ensures a clear mapping of positional information between the model's inputs and outputs, which improves the model's grounding capability.

Table 2: Results of SpiritSight on AitW, Odyssey, GUIAct(web-multi) and ScreenSpot. Data with underscores indicates different settings, where MiniCPM-GUI was not tested on the *general* part of AitW and SeeClick split the train-test set in a custom way.

| GUI Agent | Model Size | AitW AMS | Odyssey AMS | GUIAct | | ScreenSpot | | |
|---|---|---|---|---|---|---|---|---|
| | | | | TypeEM | CliACC | Web | Mobile | Desktop |
| CogAgent (Hong et al., 2024) | 18B | **76.9%** | - | - | - | 49.5% | 45.5% | 47.1% |
| SeeClick (Cheng et al., 2024) | 9.6B | 59.3% | - | - | - | 44.1% | 65.0% | 51.1% |
| OdysseyAgent (Lu et al., 2024) | 9.6B | 73.2% | 74.3% | - | - | - | - | - |
| MiniCPM-GUI (Chen et al., 2024b) | 3B | 58.4% | - | 67.0% | 47.5% | - | - | - |
| SpiritSight-2B | 2B | 72.1% | 72.3% | 67.9% | 50.2% | 63.6% | 62.5% | 61.8% |
| SpiritSight-8B | 8B | 73.6% | **75.8%** | **72.3%** | **54.6%** | **68.3%** | **68.4%** | **62.9%** |

# 5 SETTINGS

## 5.1 IMPLEMENTATION DETAILS

We use InternVL(2B, 8B and 26B) (Kim et al., 2022; Chen et al., 2024c) as pre-trained models. The history actions are limited within 5 actions to avoid excessive overload. The training process is divided into two phases: continual pre-training and fine-tuning. During the pre-training phase, we train all the collected datasets mentioned in the section 3 simultaneously. Different prompts are designed for different training tasks to avoid task confusion. We unfreeze the visual encoder, decoder, and MLP layer of InternVL. The learning rate is set to 1e-4, 1e-4, 5e-5 for 2B, 8B, 26B, respectively, and the batch size is 1024.

After pre-training, we fine-tuning our model in several downstream datasets separately. for the ScreenSpot benchmark, we follow the data proportions from Cheng et al. (2024), using part of the first-level and second-level data, as well as data from Li et al. (2020); Deka et al. (2017); Wang et al. (2021) to train the entire model. For other GUI navigation benchmarks, we first train the entire model for 1 epoch using third-level data and the training data corresponding to each benchmark, then fine-tune the model for 1 epoch on the benchmark-specific training data using LoRA (Hu et al., 2021). While training the entire model, the learning rate is set to the same as pre-training, and the batch size is 1024. During fine-tuning, the learning rate is set to 5e-5, the batch size is 64, with the alpha of visual encoder and decoder set to 32 and 64, respectively.

## 5.2 BENCHMARK & METRIC

To access SpiritSight's capability in diverse real-world environments, we evaluate SpiritSight on AitW (Rawles et al., 2024), Mind2Web (Deng et al., 2024) ScreenSpot (Cheng et al., 2024), GUIAct(web-multi) (Chen et al., 2024b), and GUI-Odyssey (Lu et al., 2024). For AitW, we use the *standard* setting for splitting training and test data and remove all the test data from pre-training set to prevent data leakage. *Action matching* is selected as the metric. For Mind2Web and ScreenSpot, we use the same process and evaluation methods as SeeClick (Cheng et al., 2024) chose. For GUI-Course, we evaluate SpiritSight in the *web-multi* data and report *Step SR* metric. For GUI-Odyssey, we report the *action matching score(AMS)*. Refer to Appendix D.1 for more information about the benchmark.

# 6 EXPERIMENT

## 6.1 ADVANCED VISUAL-BASED GUI AGENT

We evaluate SpiritSight on Mind2Web (Deng et al., 2024) benchmark, which provides high-quality and multi-dimensional test data. We compare the results of SpiritSight with other advanced methods across various input modalities and test configurations, as shown in Table 1. Methods that using top-50 candidates as input perform the best. This is evident, as the assistance of candidate elements

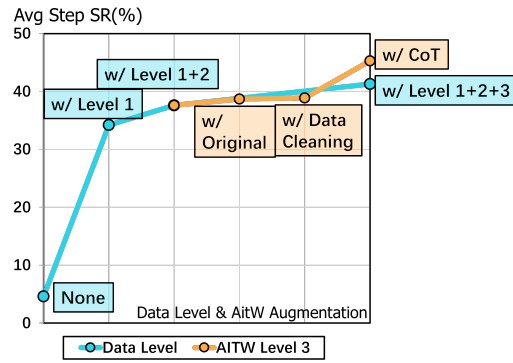

(a) Ablation of three data levels and our AitW data augmentation.

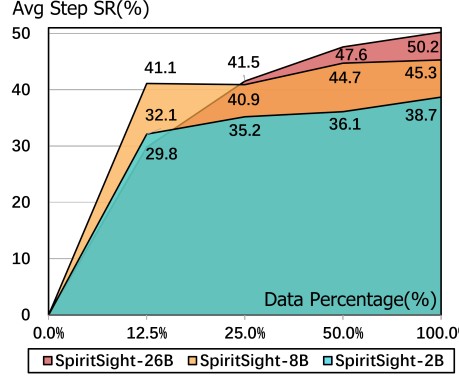

(b) Ablation of data percentages used in training.

Figure 4: The Average Step Success Rate (Avg Step SR) from the Mind2Web benchmark is used as an indicator. (a)The blue: each level of data contributes to improving Step SR. The orange: Both cleaning the AitW data and training in a CoT manner effectively improve the model's GUI navigation capabilities. (b)The performance improves as the dataset size increases. SpiritSight-26B appears to have further potential for improvement.

can significantly reduce the decision space. However, such methods are not particularly feasible in practice.

It is indicated that SpiritSight significantly outperforms all methods that do not rely on candidate elements as input, including visual-based methods, language-based methods, and even visual-language-based methods. This demonstrates strong capabilities of SpiritSight in Web GUI navigation tasks. It is noticed that SpiritSight achieved a significant advantage in the Ele.Acc metric compared to other visual-based methods, which can be attributed to the specially constructed visual grounding training data and the proposed UBP approach. We also evaluate the text grounding ability of SpiritSight on our custom text2bbox datasets. See Appendix D.2 for more details.

## 6.2 STRONG CROSS-PLATFORM COMPATIBILITY

We evaluated SpiritSight on other benchmarks across various GUI platforms and compare it with advanced visual-based Agents as shown in Table 2. SpiritSight demonstrated leading performance on most benchmarks. For ScreenSpot, a functional grounding benchmark, SpiritSight performed well across all three platforms. This not only highlights SpiritSight's cross-platform capabilities but also indicates that improving grounding enhances its GUI navigation abilities. It is noticed that SpiritSight does not perform as well on mobile platforms(AitW, Odyssey, ScreenSpot-mobile) as it does on web platforms(GUIAct, ScreenSpot-web), especially on the AitW benchmark. There are two possible reasons for this: (1)the AitW test dataset contains some annotation errors (Zhang et al., 2024a; Chai et al., 2024); (2) The dynamic resolution method may not significantly benefit navigation tasks on mobile screens due to their inherently lower information density.

## 6.3 RECOGNITION AND GROUNDING AS PRIORS FOR GUI NAVIGATION

To verify the significance of the three levels of data, we progressively removed the third-level, second-level, and first-level data from the training set during the pre-training phase. The results is shown in Figure 4a. It can be seen that each level of data contributes to improving the Step SR. While the first-level task differ the most from web navigation compared to the other two levels, they provide an effective initialization for the pre-trained model. Although the third-level data is constructed from the mobile environment, it also aids in web-based GUI navigation tasks. This indicates that the joint learning strategy helps SpiritSight develop strong navigation abilities across different GUI environments with limited resources. We also conducted ablation experiments to evaluate the effectiveness of data cleaning and CoT construction on the third-level data, as shown

Table 3: Results on GUIAct(web-multi) with different language training datasets.

| SFT Data | Overall Step SR | Chinese Step SR | English Step SR |
|---|---|---|---|
| English+Chinese | 49.3% | 49.3% | 49.2% |
| English | 35.0% | 24.5% | 48.6% |

in Figure 4a. It can be observed that both cleaning the AitW data and training in a CoT manner effectively improve the model's GUI navigation capabilities.

## 6.4 BETTER GROUNDING ABILITY FROM UBP

To verify the effectiveness of UBP on grounding task, we employ LoRA for resource efficiency to train InternVL with the same data as SeeClick (Cheng et al., 2024)in 4 different settings, and then evaluate it on Mind2Web benchmark. As shown in Figure 3c, it can be seen that UBP shows a significant improvement in Ele.Acc compared to the baseline, while the difference in Op.F1 is not substantial. This indicates that UBP improves the performance of GUI Agent primarily by enhance the grounding ability. Finally, the combination of UBP and 2D-BPE achieves the best results.

## 6.5 SCALING EFFECTS ON DATASET AND MODEL SIZE

We explored the impact of dataset and model size on SpiritSight using Mind2Web benchmark. train the entire model for 1 epoch using third-level data and Mind2Web training set. and the results are shown in Figure 4b. SpiritSight-2B, trained on just 1/8 of the dataset, achieved 32.1% Step SR, surpassing SeeClick (Cheng et al., 2024). This impressive performance comes from the high quality and grounding-focus of the collected data. The performance of the model improves as the dataset size increases, demonstrating the significance of collecting large-scale data. SpiritSight-2B reaches saturation with a smaller amount of data, while SpiritSight-26B appears to have further potential for improvement, which aligns with the scaling law of LLM.

We also tested the sensitivity of models trained on 100% of the pre-training data to downstream training data. It was noted that SpiritSight, which had not been pre-trained on web navigation data, achieved 36.6% step SR with only 1/8 of the training data, showing strong foundational capabilities in the web GUI domain.

## 6.6 EFFECTIVE TRANSFER TO OTHER LANGUAGES

Exploring the cross-lingual capabilities of GUI agents is highly beneficial for their application in non-English environments. We split the training and test sets of GUIAct(web-multi) dataset into English and Chinese parts, respectively. We then fine-tune SpiritSight-8B on two sets of data: the entire training set (English + Chinese) and the English-only training set. The results are shown in Table 3.

Under the *English + Chinese* configuration, SpiritSight achieved very similar results on both the English and Chinese test sets. Notably, SpiritSight, fine-tuned only on the English training set, achieved an Step SR of 24.5% on the Chinese test set, reaching half of the English + Chinese performance. The zero-shot capability of SpiritSight in Chinese comes from the small but effective foundational Chinese data included in the pre-training phase.

This experiment provides a paradigm for applying GUI agents to non-English environments: by collecting (1) free web and mobile GUI information from the target language environment (level 1 & level 2 data), and (2) a small amount of high-quality GUI navigation data at minimal cost (level 3 data). With this, the same capabilities as in the English environment can be achieved through training.

486
487
## 7 LIMITATION AND FUTURE WORK

488
489
490
491
**Safety and compliance issues**. As the SpiritSight Agent is a visual-based GUI agent, it constantly requires access to screenshots which may contain personal information or other sensitive data. Users and system providers should manage the system privileges granted to the SpiritSight Agent carefully to mitigate potential privacy and security risks.

492
493
494
495
496
497
498
**Increased computational demands with higher input resolutions**. The computational requirements of the SpiritSight Agent increase with the resolution of the input images. The inference latency for each step will get longer as the input image size get larger. Yet, for the real-world usage of GUI Agents, the fluency of operation is pivotal to user's experience. Future work could explore more efficient model architectures or compression techniques to address these computational challenges.

499
500
## 8 CONCLUSION

501
502
503
504
505
506
507
508
In this paper, we propose an advanced visual-based end-to-end GUI agent——SpiritSight, with high generalization across multiple GUI platforms. We construct a efficient mutli-level, large-scale, high-quality GUI pre-training data to equip SpiritSight with robust GUI perception, grounding and understanding capabilities. We further introduce UBP method to resolve the ambiguity in Dynamic High-Resolution input during model training, further enhancing the ability of SpiritSight to ground GUI objects. Ultimately, SpiritSight shows strong performance in numerous GUI navigation benchmark across various GUI platforms, demonstrating great potential for practical deployment in real-world applications.

509
510
511
## 9 ETHICS STATEMENT

512
513
514
515
**Online content is uncontrollable** The screenshots in our web dataset are collected from online environments. Although we have implemented measures such as filtering for sensitive words and manual sampling checks to screen for offensive content, we still cannot guarantee that all such content has been removed.

516
517
518
519
**Computational and Energy Demands** Training large language models is a computationally intense process that requires substantial electrical power. In response to these challenges, we have designed and conducted essential experiments to optimize the training process, aiming to reduce energy consumption.

520
521
522
523
**Regulations** As GUI Agents interact with user interfaces, they access sensitive data and perform tasks that could potentially breach privacy or violate data protection laws. Ensuring these agents operate within legal frameworks is crucial to prevent unauthorized data access and misuse.

524
525
526
## 10 REPRODUCIBILITY

527
528
529
**Data Collection** We briefly explain the data collection approach in Chapter 3, and detail the specific process and nuances of the collection in Appendix F. This allows the data collection to be reproducible.

530
531
**Universal Block Parsing** We list the core formulas of UBP in Chapter 4, which are very easy to implement. Therefore, UBP is fully reproducible.

532
533
534
535
536
537
538
539

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

# A   EXTENDED RELATED WORK

## A.1   LARGE-SCALE LANGUAGE MODELS

In recent years, large language models (LLMs)  (Radford et al., 2018; Devlin, 2018; Raffel et al., 2020; Xu et al., 2021; Wu et al., 2023; Nijkamp et al., 2022; Roziere et al., 2023; Yu et al., 2023; Wang et al., 2023a; Ying et al., 2024; Shao et al., 2024; Wei et al., 2022b;a; Pan et al., 2023) have demonstrated remarkable capabilities in the field of Natural Language Processing (NLP), encompassing natural language generation, commonsense knowledge question-answering, code completion, mathematical computation, and logical reasoning.  LLM have also demonstrated strong decision-making capabilities, laying the foundation for the emergence of GUI agents.

## A.2   MULTI-MODAL LARGE LANGUAGE MODELS

With the development of large language models, numerous works  (Bai et al., 2023; Wang et al., 2023b; Lin et al., 2023; Li et al., 2024; Chen et al., 2024c; Zhang et al., 2024b; Yao et al., 2024; Jin et al., 2023) have proposed Multi-modal Large Language Models (MLLMs) to bring the capabilities of language models into the visual domain. CLIP  (Hafner et al., 2021) uses contrastive learning to align visual and language features, while BLIP  (Li et al., 2022) and BLIP-2  (Li et al., 2023)build on this by adding a language decoder, enabling the models to perform image-grounded text generation. InternVL  (Chen et al., 2024d) attempts to scale the parameters of visual encoder up to 6 billion, significantly enhancing the model's ability to perceive visual input. LLaVA  (Liu et al., 2023) and Sphinx  (Lin et al., 2023) improve the models' understanding and chat abilities through instruction tuning and multitask learning, respectively. Beyond general domains, OCR-Free  (Kim et al., 2022) methods use an encoder-decoder architecture to achieve end-to-end visual document understanding. This demonstrates the significant potential of MLLM in GUI navigation task.

# B   TASK FORMULATION

For a given GUI platform, we first obtain an action space $\mathcal{A}$ which contains all possible action that an agent can take. Given the task description $\mathcal{T}$, the previous actions $\mathcal{H} = \{a_1, a_2, ..., a_{t-1}\}$, the action space $\mathcal{A}$ and the current screenshot $o_t$, the agent is expected to infer the optimal action $a_t^*$ that maximizes the expected future reward. The inference process is guided by a policy $\pi$, as shown below, which maps the current context to a probability distribution over the action space $\mathcal{A}$.

$$a_t^* \sim \pi(a|\mathcal{T}, \mathcal{H}, \mathcal{A}, o_t) \tag{3}$$

We propose a hierarchical decomposition of the policy to handle the complexity of action inference. Initially, the overall policy $\pi$ is decomposed into step inference policy $\pi_s(s|\mathcal{T}, \mathcal{H}, o_t)$ and action inference policy $\pi_a(a|s, \mathcal{A})$. The step inference policy $\pi_s$ selects the step $s$ based on the current context, where the step is defined as the natural language description of the action. Once the step $s$ is determined, the action inference policy $\pi_a$ selects a specific action $a$ from the action space $\mathcal{A}$ conditioned on $s$.

Further, we decompose $\pi_a$ into two additional sub-policies: $\pi_{pos}(a_{pos}|s, \mathcal{A})$ and $\pi_{attr}(a_{attr}|s, \mathcal{A})$. Here, $a_{pos}$ corresponds to the positional information of the action, typically the coordinates where the action is performed, while $a_{attr}$ represents the non-positional information of the action, such as the action type (*c*lick, input) or additional parameters like input text. The decomposition is formally expressed as:

$$\pi(a|\mathcal{T}, \mathcal{H}, \mathcal{A}, o_t) = \pi_s(s|\mathcal{T}, \mathcal{H}, o_t) \cdot \pi_{pos}(a_pos|s, \mathcal{A}) \cdot \pi_{attr}(a_{attr}|s, \mathcal{A}) \tag{4}$$

It is easy for visual-based agent to learn on the step inference policy $\pi_s$, as modern VLLMs performs well on reasoning and decision-making. The non-positional inference policy $\pi_{attr}$ is also easy since the non-positional part of a action can be directly paraphrased according to the step. For example, *INPUT("Copenhagen")* can be directly infer through the step *Input "Copenhagen" into the arrival input box*. The primary challenge lies in learning the positional sub-policy $\pi_{pos}$ as mentioned in Chapter 2. Based on this, we construct a large scale dataset with a primary focus on grounding tasks to address the challenge of learning accurate positional actions.

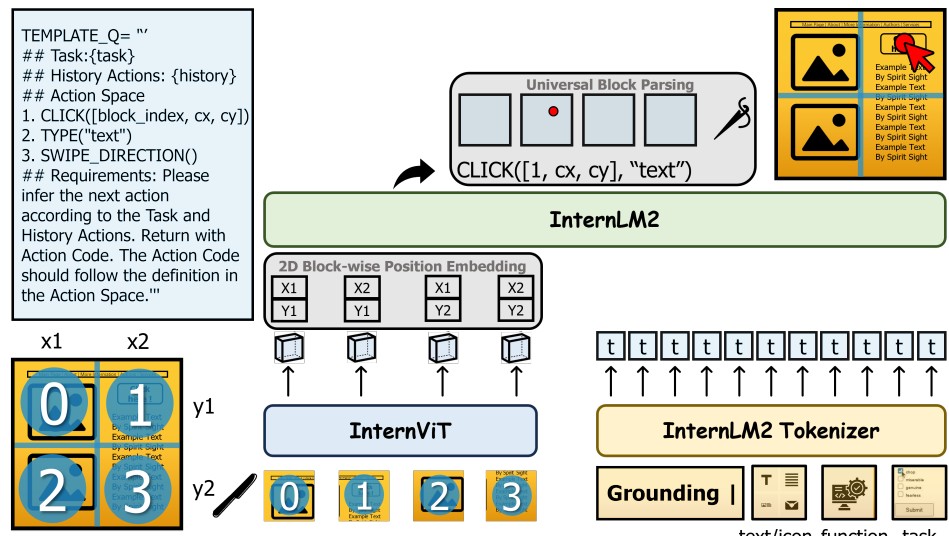

Figure 5: The overall architecture of SpiritSight. SpiritSight is pre-trained on large-scale, multi-level, high quality datasets. The UBP solve the ambiguity in Dynamic High-Resolution input during model training.

## C  OVERALL ARCHITECTURE

We build our model based on the pre-trained InternVL2.0 (Chen et al., 2024c) (InternVL for short), a family of advanced and open-sourced VLMs. We chose InternVL for the following reasons: (1) The large-scale and high-performance visual encoder is more capable to handle the text-rich GUI environment. (2) The dynamic resolution strategy largely preserves the details of the input screenshots, allowing for the perception of fine-grained text and icon information. We take the advantage of large-scaled visual encoder with a large-scaled GUI dataset described in chapter 3. We further propose a Universal Block Parsing (UBP) method to handle with the small object localization problem brought by dynamic resolution in chapter 4.

The architecture of SpiritSight is depicted in Figure 5. To begin with, the input image is the GUI screenshot. According to the dynamic resolution algorithm of InternVL, the appropriate ratio of input image is decided. Then, the image is divided into several blocks, each with a unique index, in preparation for the post-processing phase of our UBP method. These image blocks will be flattened as batches before they are sent into visual encoder, which results in the loss of their 2D spatial relation. To address this issue, we introduce the 2D Block-wise Position Embedding (2D-BPE)(Kim et al., 2022) method, which maintains the blocks' 2D spatial relation by adding a row embedding and column embedding to each block. Afterwards, the embedded image features, along with the task objective, the action space and the history actions are passed through the InternLM2 decoder to make the action code inference. Finally, the exact pixel coordinate and the action to be executed is obtained by the UBP parser. We define a separate action space $\mathbb{A}_{space}$ for each GUI platforms, making our SpiritSight model highly compatible to a variety of GUI navigation tasks. The history actions H are limited within 5 actions to avoid excessive history overload. For each step, SpiritSight would observe the current screen, output the optimal action-code A according to task objective T, history actions H and the given action space $\mathbb{A}_{space}$. The detailed prompt template can be seen in Appendix H.

## D  EXTENDED EXPERIMENTS

### D.1  GUI AGENT BENCHMARK

In recent years, GUI agents have seen rapid development, with many types of benchmarks emerging. MiniWoB (Shi et al., 2017), MiniWoB++ (Liu et al., 2018), and WebShop (Yao et al., 2022) are

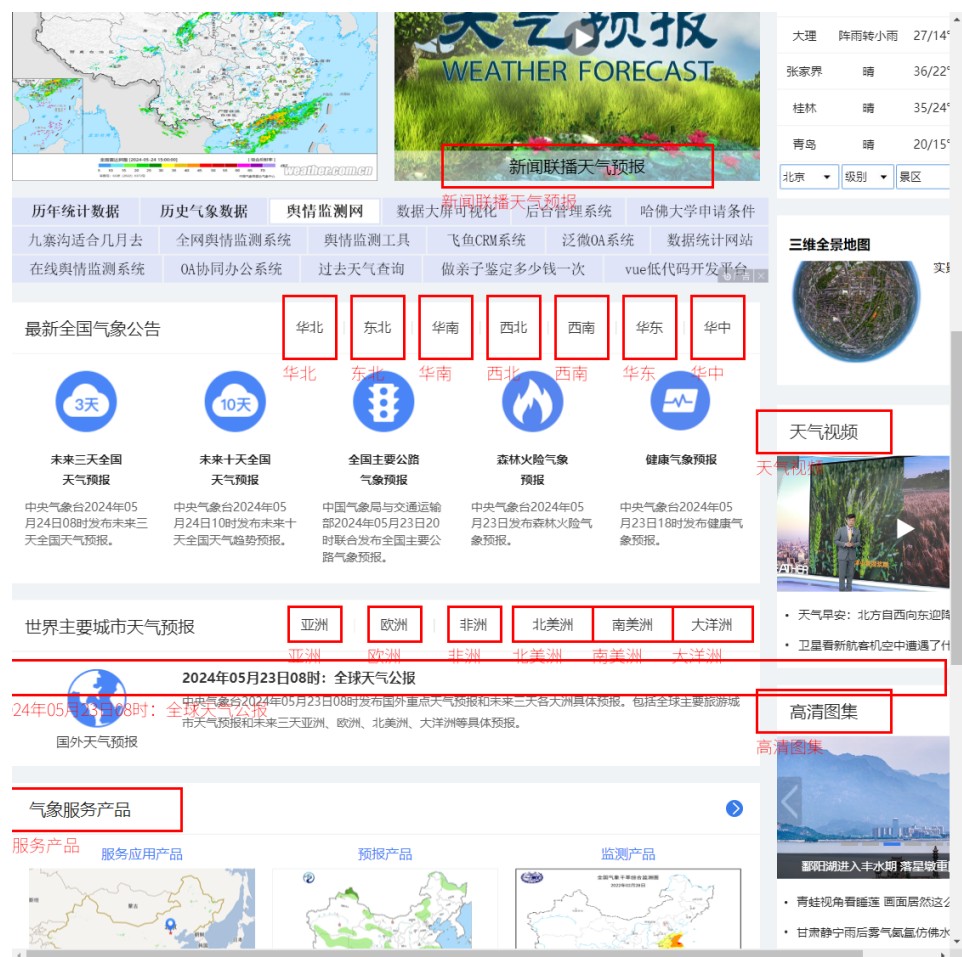

Figure 6: Visualization results of SpiritSight-2B on our custom text2bbox test set. we select a Chinese web page to show the cross-lingual capabilities of our models. The red boxes represent the generated results and the text next to it represent the text prompt.

early classic GUI navigation benchmarks. However, the data in these benchmarks is synthetically generated, which creates a slight gap compared to real-world data. AitW Rawles et al. (2024) is a large real-world dataset that is currently popular for mobile GUI navigation. Mind2Web Deng et al. (2024) is a benchmark for web navigation that has become representative due to its high quality and its provision of cross-task, cross-website, and cross-domain evaluations. ScreenSpot Cheng et al. (2024) is a benchmark for functional grounding, covering mobile, web, and desktop scenarios. GUIAct (Chen et al., 2024b) and GUI Odyssey (Lu et al., 2024) are newly released benchmarks designed for web and mobile environments, respectively. They rely on human annotations, and the annotations underwent quality checks, making them highly reliable benchmarks.

## D.2 GUI GROUNDING ABILITIES

To evaluate SpiritSight's ability in text localization, we construct a small text2bbox benchmark. We select a small number of URLs from the website URL mentioned in Appendix F. These URLs are not included in the training set. Following the method described in Chapter 3, we construct a text2bbox task, resulting in 3,700 text2bbox pairs as the test set. We adopt the same metric as in SeeClick (Cheng et al., 2024), where the goal is to determine whether the predicted center point falls within the ground-truth bounding boxes. Finally, SpiritSight-2B achieves a 96.1% hit rate on this test set, demonstrating its strong capability in fundamental grounding tasks. Fig. 6 shows the

visualization of the predicted bounding boxes from SpiritSight-2B, where we select a Chinese web page to show the cross-lingual capabilities of our models.

## E  BBOX2DOM TASK EXAMPLE

Figure 7: An example of the Bbox2dom task. Left shows a given bounding box on a web page, right shows its corresponding simplified DOM structure.

## F  DATA COLLECTION

### F.1  WEB DATA COLLECTION

We collected website URLs from two sources: the Common Crawl (Group, 2024) datasets and URLs from website ranking. We then developed a data collection tool using playwright (Microsoft, 2024) library to get real-world web data from the collected URLs. We used the URLs from the website ranking as a supplement to Common Crawl due to their compromised quality, including a large proportion of blank pages, sparse-texted pages, and dead pages. We developed a data collection tool using playwright (Microsoft, 2024) library to get real-world web data from the collected URLs.

We sequentially traverse the collected URLs. For each URL, we start data collection only after the page has fully loaded. The data collected includes website screenshots and hierarchical element structure information. A carefully designed scheme is used to collect the element hierarchy. First, we perform grid sampling on the coordinates, with a step size of 8, to ensure that elements with a length or width greater than 8 are captured. Then, we obtain the corresponding element objects based on the sampled coordinates, which include various information about the elements, such as type, inner text, coordinates, and interactivity. We determine whether an element is clickable by checking its pointer property and registered events, and assess whether it can accept text input by checking its type. We also label these element objects to simplify the HTML information. Specifically, after collecting all the element information, we developed an HTML pruning algorithm to simplify the HTML structure. Through this pruning algorithm, all labeled element nodes and those with structural representation functions are retained. The resulting DOM trees are used to construct the bbox2dom data.

After collecting the data from the current website, we acquire new pages using two methods: scrolling down or clicking on an element, with clickable elements randomly sampled from all available ones. We collect 30 pages for each initial URL. This process is repeated, creating a continuous cycle of interaction. Ultimately, we collect 740k web-page screenshots along with their DOM annotation. Among them, English websites account for 3/4, while Chinese websites account for 1/4.

## G  TRAINING DATA FORMAT

We constructed a large number of text2bbox, bbox2text, bbox2dom, and function2bbox tasks. Each sample contains multiple data pairs to fully utilize the context length and improve the efficiency and stability of the model during training. It is worth noting that we adopted a representation with an attached block index, which is derived from our proposed UBP method. Below are the training data templates for each task, where the prompts used in the actual data construction are not fixed but randomly selected from a prompt pool.

***Data Format for text2bbox Task***
*user:*
<image>
1.{text 1}
2.{text 2}
3.{text 3}
...
Provide the bounding boxes of each given text in a list format.
*assistant:*
1.{[block-index, cx, cy, w, h]}
2.{[block-index, cx, cy, w, h]}
3.{[block-index, cx, cy, w, h]}
...

***Data Format for bbox2text Task***
*user:*
<image>
1.{[block-index, cx, cy, w, h]}
2.{[block-index, cx, cy, w, h]}
3.{[block-index, cx, cy, w, h]}
...
Provide the text content of each given bounding box in a list format.
*assist:*
1.{function description 1}
2.{function description 2}
3.{function description 3}
...

***Data Format for bbox2dom Task***
*user:*
<image>
I'd like some information about the specific region [block-idx, cx, cy, w, h] in the image.
*assistant:*
{DOM_Tree}

***Data Format for function2bbox Task***
*user:*
<image>
1.{function description 1}
2.{function description 2}
3.{function description 3}
...
In this image from a webpage, find out where to click for a certain need and provide bbox coordinates in a list format.
*assistant*
1.{[block-index, cx, cy, w, h]}
2.{[block-index, cx, cy, w, h]}
3.{[block-index, cx, cy, w, h]}
...

## H  PROMPT TEMPLATES

### H.1  EVALUATION INFERENCE

> **Prompt for Evaluation Inference**
> ## Task: {task}
> ## History Actions:
> {history}
> ## Action Space
> {Action Space}
> ## Requirements: Please infer the next action according to the Task and History Actions.
> Return with Action Code. The Action Code should follow the definition in the Action Space.

### H.2  LEVEL-TWO FUNCTION GENERATION

> **Prompt for Level-two Function Generation**
> Please infer the purpose of the operation "click on the '{text}' on the {region} of the webpage" based on the webpage.
> Please deliver the purpose specifically and clearly, which points to the certain item.
> Its direct context includes the following information: {context_text}.
> Please make the answer only in English.
> Let's think step by step.
> Your final answer should be in a new line and included in double quotation like:
> The purpose is "xxx".

> **Prompt for Level-two Function Augmentation**
> Can you rewrite the original purpose "{purpose}" into a short phrase?
> Here are some examples:
> {Few-shot example 1}
> {Few-shot example 2}
> {Few-shot example 3}
> Output only the refined purpose, start with 'to', without any explanation.

### H.3  LEVEL-THREE DATA PROCESSING

> **System Prompt for Level-three Data Processing**
> You are a mobile operation assistant, the main goal is to help identify whether the mobile navigation operation is correct.

> **Prompt for Level-three Single Step Data Processing**
> Task: {task}
> Action History: {history}
> The Next Action: {action}
> Return:
> 1. Summarize the screenshot of a mobile phone about its main content and its functionality. Describe it with necessary details, but not too long.
> 2. Based on the task, action history, the current screen and your summary, estimate the purpose of the next action. Note that it is not the entire goal, but a single step goal for the next step. Return only the purpose.
> 3. Analyze the rationality of the next action. Return with the reason.
> 4. Return the final answer of the rationality with just 'True' or 'False'.

**Prompt for Level-three Multi Step Data Processing**
Task: {task}
Action History: {history}
The Current Action: {action}
You are completing a mobile task and now in step {step_idx}. Picture 1 shows the current screen with action demonstration and picture 2 shows the screen after performing The Current Action on picture 1. You are also given the Action History before the Current Action.
Return:
1. Summarize picture 1 about its main content and its functionality. Also describe the changes that have occurred in Figure 2 compared to Figure 1. Describe them with necessary details, but not too long.
2. Based on the changes between Figure 1 and Figure 2, estimate the function of the Current Action. Return with format of "The function of the Current Action: xxx"
3. Analyze the rationality of the Current Action based on the Task. Return only the reason.
4. Return the final answer of the rationality of the Current Action with just 'True' or 'False'.
5. Analyze if the Task is successfully completed. Return only the reason.
6. Return the final answer of the complementarity of the Task with just 'True' or 'False'.

---

**Prompt for Level-three Multi Step Marked Data Processing**
Task: {task}
Action History: {history}
The Current Action: {action}
You are completing a mobile task and now in step {step_idx}. Picture 1 shows the current screen with action demonstration and picture 2 shows the screen after performing The Current Action on picture 1. You are also given the Action History before the Current Action.
Return:
1. Describe {mark}.
2. Summarize picture 1 about its main content and its functionality. Also describe the changes that have occurred in Figure 2 compared to Figure 1. Describe them with necessary details, but not too long.
3. Based on the changes between Figure 1 and Figure 2, estimate the function of the Current Action. Return with format of "The function of the Current Action: xxx"
4. Determine the Current Action is "Click" or "Long Press" based on the previous information. (The Current Action: xxx)
5. Analyze the rationality of the Current Action based on the Task. Return only the reason.
6. Return the final answer of the rationality of the Current Action with just 'True' or 'False'.
7. Analyze if the Task is successfully completed. Return only the reason.
8. Return the final answer of the complementarity of the Task with just 'True' or 'False'.

---

**Prompt for Level-three Last Step Data Processing**
Task: {task}
Action History: {history}
You have just completed a mobile task with a series of actions listed in Action History. The picture shows the final screen of the mobile.
Return:
1. Summarize the picture about its main content and its functionality. Describe it with necessary details, but not too long.
2. Analyze if the task is successfully completed from the perspectives of success and completion separately.
3. Return the final answer of the analysis with just 'True' or 'False'.

