# OpenReview forum: "SpiritSight Agent: Advanced GUI Agent with One Look"
_ICLR.cc/2025/Conference — ICLR 2025 Conference Withdrawn Submission_

### Official Review · Reviewer_qbMB · 2024-11-02

**Soundness:** 3
**Presentation:** 2
**Contribution:** 2
**Rating:** 5
**Confidence:** 4

**Summary:**

This paper tackles challenging GUI understanding and planning tasks by proposing a VLM-based GUI agent trained with specially curated data and improved input-output designs. Specifically, the authors annotated a large amount of GUI training data to enhance the GUI understanding capabilities. To solve the ambiguous element grounding issue caused by the image-blocking approach commonly used in existing VLMs like InterVL, the authors introduced block-wise positional embedding and a novel block-specific coordinate output style to improve element grounding accuracy. The experiments show that the proposed methods achieves higher grounding accuracy and planning success rates.

**Strengths:**

1. The authors curated a large GUI understanding dataset, which may be a contribution to the community.
2. The authors recognized the ambiguity in the blocking method used in existing VLMs, which is insightful.
3. The experiments are comprehensive.

**Weaknesses:**

1. The authors failed to provide experimental details of human evaluation (mentioned in L233) to verify the quality of data, which makes the VLM-aided data annotation pipeline seem unreliable.

2. The reasons behind incorporating the task types mentioned in L206 are stated, but no ablation experiments have been conducted to justify the selection of these task types. The authors are expected to investigate the effect of each task type to provide more solid insights.

3. The reviewer is unclear about the novelty of the curated data mentioned in Section 3. The authors are expected to compare their proposed dataset with existing datasets, such as the datasets collected in CogAgent and SeeClick; otherwise, the authors' data collection can hardly be regarded as a novel contribution.

4. Unclear experimental analysis. L415 states "This indicates that improving grounding enhances its GUI navigation abilities". The reviewer doubts that the experiments are sufficient to justify this argument. If the authors hold this argument by comparing SpiritSight with existing methods, such as SeeClick, this argument is not solid enough because the base VLMs are different, and the downstream GUI navigation fine-tuning data are also different.

**Questions:**

1. How to interpret "SpiritSight does not perform as well on mobile platforms(AitW, Odyssey, ScreenSpot-mobile) as
it does on web platforms(GUIAct, ScreenSpot-web)" in L416? Table 2 shows that the performances on ScreenSpot-mobile and ScreenSpot-web are almost the same.

---

### Official Review · Reviewer_MRfP · 2024-11-03

**Soundness:** 3
**Presentation:** 2
**Contribution:** 2
**Rating:** 5
**Confidence:** 4

**Summary:**

This paper introduces a novel visual-based GUI agent that excels in navigation tasks by leveraging a large, cost-effective dataset and a Universal Block Parsing (UBP) method for enhanced element grounding and localization. Empirical results demonstrate that the proposed method outperforms existing baselines, particularly in text/icon grounding and GUI navigation, while offering scalable versions to meet different computational needs.

**Strengths:**

1. The paper proposes a new cost-effective GUI dataset of 5.46 million samples and introduces a Universal Block Parsing (UBP) method to resolve the ambiguity.
2. The proposed method was robust, and the performance is promising, showing the effectiveness of the designs.
3. The writing of this paper is easy to follow.

**Weaknesses:**

1. The experimental conclusions are not very rigorous. For example, in Lines 414-415, how do the results on ScreenSpot in Table 2 demonstrate that improving grounding further enhances navigation ability?
2. The lack of ablation experiments fails to demonstrate the effectiveness of the UBP module.
3. Lack of qualitative or quantitative analyses to assess the reliability of the cleaning process (e.g., Section 3.3).

**Questions:**

1. The process of cleaning the AitW dataset (Section 3.3) is not entirely clear. If GPT-4o determines a particular step is unreliable, is the entire trajectory discarded?

---

### Official Review · Reviewer_grGR · 2024-11-04

**Soundness:** 3
**Presentation:** 3
**Contribution:** 2
**Rating:** 6
**Confidence:** 4

**Summary:**

In this paper, the authors propose SpiritSight which advances automated GUI navigation by introducing a visual-only approach that eliminates the need for HTML/XML access. At its core is the Universal Block Parsing method, which resolves positional ambiguity by treating element localization as a multi-image QA problem, supported by a hierarchically structured dataset of 5.46 million samples. The system demonstrates exceptional performance across platforms and languages.

**Strengths:**

- The authors focus on a visual-based GUI agent approach that avoids obtaining UI source code data, relying solely on screenshots.

- This paper introduces Universal Block Parsing, an innovative solution to address positional ambiguity in dynamic resolution approaches.

- In this paper, the authors propose a hierarchical data collection strategy that offers a practical, scalable approach to GUI agent development.

**Weaknesses:**

- The authors claim SpiritSight is a generalist agent with cross-platform capability. However, the evaluation methodology involves fine-tuning the model separately on each downstream dataset. This reliance on scenario-specific fine-tuning suggests that the model's strong performance is not due to inherent generalization but rather tailored adjustments for each platform, which undermines the assertion of cross-platform generality.
- The ablation studies are somewhat limited. The author only provides basic comparisons with and without UBP and 2D-BPE, lacking analyses such as the impact of different block sizes on performance and how UBP manages elements that cross block boundaries.

**Questions:**

- Does the block size have to be the same as the patch size of the ViT encoder?
- Have you examined the impact of reducing grounding precision from pixel-level to block-level?

---

### Official Review · Reviewer_J47g · 2024-11-06

**Soundness:** 2
**Presentation:** 2
**Contribution:** 2
**Rating:** 5
**Confidence:** 4

**Summary:**

The work introduces SpiritSight, a visual-based generalist end-to-end GUI agent designed to enhance human-computer interaction by automating navigation on digital devices. It proposes a curriculum learning framework to train the model with robust GUI understanding and localization capabilities. SpiritSight also employs a Universal Block Parsing (UBP) method to enhance the model's ability to correspond to GUI objects, thereby avoiding ambiguity. Experimental results across various GUI navigation benchmarks demonstrate its superior capability and compatibility.

**Strengths:**

1. The work proposes a visual-based generalist end-to-end GUI agent model for GUI grounding and action tasks and introduces curriculum training datasets for model training.
2. This work explores a novel form of modeling for GUI grounding tasks.

**Weaknesses:**

1. The SpiritSight Agent is mainly based on InternVL models while the baselines are based on various other VL backbones, such as SeeClick using QWen-VL, CogAgent are based on CogVLM. I realize that there may not be a GUI agent model using intern vl as a backbone, but how to prove that the effect is not just a boost from the VLM backbone is necessary.
2. The details of the UBP strategy are missing. The author's statement that “block slicing doesn't affect the GUI icon” requires some statistical information. Moreover, they also need to compare the UBP with other formats (e.g. coordinates) for comparison.

**Questions:**

See the Weakness part.

---

### Note · Authors · 2024-11-14

I have read and agree with the venue's withdrawal policy on behalf of myself and my co-authors.